# A National Communication Campaign in Indonesia Is Associated with Improved WASH-Related Knowledge and Behaviors in Indonesian Mothers

**DOI:** 10.3390/ijerph17103727

**Published:** 2020-05-25

**Authors:** Curtis Hanson, Emily Allen, Margie Fullmer, Rachel O’Brien, Kirk Dearden, Joshua Garn, Cut Novianti Rachmi, Jeffrey Glenn, Joshua West, Benjamin Crookston, Parley Hall

**Affiliations:** 1Department of Public Health, Brigham Young University, Provo, UT 84602, USA; curtisha.byu@gmail.com (C.H.); emily.d.allen8@gmail.com (E.A.); fullmerster@gmail.com (M.F.); rachelgoodrich15@gmail.com (R.O.); jeff_glenn@byu.edu (J.G.); josh.west@byu.edu (J.W.); benjamin_crookston@byu.edu (B.C.); 2IMA World Health, Washington, DC 20036, USA; kdearden@imaworldhealth.org; 3School of Community Health Sciences, University of Nevada, Reno, NV 89557, USA; jgarn@unr.edu; 4Fakultas Kedokteran, Universitas Padjadjaran, Jawa Barat 45363, Indonesia; cutnovianti@gmail.com

**Keywords:** WASH, media messages, interpersonal communication strategies, knowledge, behavior

## Abstract

*Background:* Water, sanitation, and hygiene (WASH) behaviors play a significant role in stunting. Knowledge and behaviors regarding WASH among caregivers are critical to providing children with chances to survive and thrive. The purpose of this study is to determine if exposure to a national communication campaign using media and interpersonal communication (IPC) is associated with WASH-related knowledge and behaviors among Indonesian mothers with children under the age of two. *Methods:* Data came from a cross-sectional survey of 1734 mothers with children under the age of two. The measures included exposure to two different interventions: media messages (media) and interpersonal communication strategies (IPC) and WASH-related knowledge and behavior. Multiple logistic regression was used to examine the association between intervention exposure and study variables. *Results:* Exposure to both media and IPC interventions was associated with participants having a higher knowledge of appropriate defecation practices (*p* < 0.001), higher knowledge of proper handwashing practices (*p* < 0.001), and higher self-reported handwashing at critical times (*p* < 0.001) but was not associated with reported practicing of appropriate defecation (OR = 0.780, 95% CI: 0.566–1.101). Mothers exposed to only media interventions were more likely to have knowledge of appropriate defecation practices (*p* < 0.001) and to have reported practicing appropriate defecation behaviors (OR = 1.539, 95% CI: 1.173–2.019). Mothers exposed to only IPC interventions were more likely to have reported handwashing at critical times (*p* = 0.009). *Conclusions:* Exposure to both media and IPC interventions was associated with increased knowledge and optimal behaviors related to WASH. These findings demonstrate the value of communications campaigns that use mass media coupled with IPC to improve WASH knowledge and behavior.

## 1. Introduction

Proper water, sanitation, and hygiene (WASH) practices are critical to the physical and mental development of children under the age of two [1]. While several systematic reviews and recent trials assessing WASH and anthropometric outcomes have found either mixed or null results [2,3,4,5], some individual studies have found that the prevention of diarrheal disease and undernutrition through proper WASH practices helps to decrease the prevalence of stunting [6,7,8,9,10,11,12,13]. Stunting has been shown to correspond with adverse health effects such as decreased cognitive and motor development, lowered performance at school, and reduced productivity in adulthood [14]. The combined effects of stunting are estimated to cost African and Asian nations up to 11% of their gross national product [15].

Limited access to WASH services presents significant challenges to preventing childhood stunting in Indonesia. Despite Indonesia’s emerging economy, the country ranks fifth highest in the world for childhood stunting [16]. Approximately 37% of children under the age of five are stunted nationwide, a percentage that increases for children in rural regions [17]. A recent study of determinants of stunting in rural Indonesia indicated that 61% of participant households had an improved sanitary facility, 43.4% reported safely disposed of a child’s feces, 55.3% used soap for handwashing, and 31.6% had an improved source of drinking water [18]. It is estimated that approximately 20% of Indonesians, or 51 million people, defecate in open spaces, such as fields, bushes, and beaches because of limited access to sanitation facilities [19].

In 2014, a social and behavior change campaign called the National Nutrition Communication Campaign (NNCC) was launched in Indonesia. The NNCC included both mass media communications (television, radio, social media) and interpersonal communication (IPC) approaches. Generally, IPC strategies include a face-to-face verbal two-way communication inclusive of listening, dialoging, and actioning [20]. In particular, IPC interventions included in the NNCC focused on health workers communicating with mothers at women’s groups, maternal health classes, and Posyandu (integrated health post) services for women and children throughout rural Indonesia. The objective of this campaign was to reduce stunting among children in Indonesia by, among other things, improving the WASH behaviors of women and children through mass media efforts and interpersonal communication (IPC). The purpose of this study is to better understand the association between exposure to these various communication interventions and their impact on WASH knowledge and behaviors. In particular, this study aimed to answer the following research questions: (1) Was exposure to NNCC media messages associated with WASH-related knowledge and behaviors? (2) Was exposure to NNCC IPC interventions associated with WASH-related knowledge and behaviors? (3) Was exposure to both NNCC media messages and IPC interventions associated with WASH-related knowledge and behaviors?

## 2. Materials and Methods

### 2.1. Design

Data for this study were collected through a cross-sectional survey conducted in rural Indonesia following the 2014–2018 NNCC intervention and represents a collaborative effort between the IMA World Health (IMA), the University of Indonesia’s Center for Nutrition and Health Studies, and the Ministry of Health in Indonesia.

### 2.2. Sample

The study sample consisted of mothers of children under the age of two from three rural districts (Banyuasin, Kubu Raya, and Katingan) located in three provinces (South Sumatera, West Kalimantan, and Central Kalimantan) in Indonesia. A multi-level sampling strategy was used to construct the study sample. One district was randomly selected from each of the three provinces. Within each of the three rural districts, 30 villages were randomly selected, and each represented a cluster unit. At a more local level, four sub-villages were randomly selected from within each of the 30 villages in each of the three districts. Finally, five mothers were selected from each sub-village. The target sample size was 600 mothers from each of the three districts; 1800 total. The actual final study sample included 1734 mothers.

### 2.3. Ethics

Ethical approval was obtained from the Ethical Research Committee at the Faculty of Public Health, Universitas Indonesia (Ref. 71/H2.F10/PPM.00.02/2018). Reconstra Utama Integra, a research firm from Jakarta, conducted the data collection. Signed informed consent was sought from each participant prior to the interview, and the participation of all subjects was voluntary. Survey data were collected using an electronic tablet by experienced interviewers and field coordinators. Each interviewer interviewed approximately six respondents per day and reported to field coordinators, who then verified the responses and uploaded survey data daily. A data manager checked data and noted any errors. De-identified data were received by the authors from Reconstra Utama Integra to perform the analyses.

### 2.4. Data Collection

Posyandu, the community-based health outpost, was the primary site for IPC interventions. From this facility, participating mothers received postnatal care, growth monitoring and health promotion sessions that were held once monthly and focused on (1) nutrition (maternal, infant, and young child, including breastfeeding, and complementary feeding) and (2) sanitation and hygiene (latrine usage and handwashing). The mothers’ participation in IPC programming was completely voluntary.

### 2.5. Measures

Interviewers collected data on participants’ demographic information, media and IPC exposure, and WASH knowledge and practices. Demographic information was collected and included the age of the mother and child, the mother’s education level, and the total household income level, as represented in Indonesian Rupiah (IDR). To ascertain media exposure, interviewers showed respondents a brief video clip of the TV commercials, or an image of the print media or social media page and asked if they had seen the particular media. Exposure was confirmed by asking the respondent to describe the theme or message in the commercial, print media, or social media. Respondents were considered to have been exposed to the media component if they were able to accurately describe the theme or message of the ad in its various platforms. Exposure to the IPC component was determined by asking respondents if they had participated in classes or support groups for mothers of children under the age of two who meet regularly to share experiences, discuss, and give support for mother and child’s health primarily related to pregnancy, breastfeeding, and nutrition which was facilitated by the Posyandu (an Indonesian community-based health outpost). If respondents said yes, they were asked to provide a description of the topics of the meetings. Respondents were considered to be exposed to the IPC component if they were able to accurately describe the topics of the meetings.

Two items were created to represent the participants’ knowledge of WASH. A first composite variable was constructed using the participant responses from three questions related to open defecation—the ability to correctly identify (1) at least one of the risks of open defecation (e.g., the transmission of germs, causing diarrhea), (2) at least one medium by which germs could be transmitted to a child from human feces (e.g., fly, water, dirt), and (3) that defecation should occur in a hygienic latrine/toilet (yes/no). A second composite variable was created using participant responses from three questions relating to handwashing—the ability to correctly (1) report the steps involved in proper handwashing (e.g., with soap and clean water), (2) identify the appropriate times for handwashing (e.g., after defecation, before preparing meals, after cleaning a baby), and (3) identify the meaning of CTPS—“*Cuci Tangan Pakai Sabun*” (handwashing with soap movement). Correct responses to each question resulted in a value of 1. Each composite variable was the summation of the values for the three corresponding questions. This resulted in four categories for both composite variables: no knowledge, low knowledge, medium knowledge, and high knowledge.

Participants were also asked about two WASH-related behaviors. First, participants reported the extent to which they engaged in appropriate (vs. inappropriate) defecation. Appropriate defecation was defined as using a gooseneck toilet, squat toilet with no floor, or squat toilet with floor, and discarding feces in a septic tank or a closed ground hole. Second, participants reported their handwashing behaviors at critical times, which included after defecation, after cleaning a baby, before preparing meals, before eating meals, and before breastfeeding. A frequency measure (0–5) was used to reflect the extent to which they engaged in handwashing at critical times (e.g., never, at one time, two times, three times, four times, or at every critical time).

### 2.6. Analysis

The SAS software version 9.4 (SAS Institute; Cary, NC, USA) was used to conduct all analyses. Demographic data were described using basic frequency statistics. Both linear and logistic adjusted regression models were run for four key dependent variables: (1) knowledge of defecation, (2) knowledge of handwashing, (3) appropriate defecation behavior, and (4) handwashing behavior. Primary independent variables were exposure levels to IPC, media, or both. All models controlled for mother’s education, mother’s age, and total household income.

## 3. Results

Respondent demographics are presented in Table 1. The mean age for respondents was 27.72 years. Most participants had at least some education, with primary school (38.64%), junior high school (24.39%), and senior high school (25.03%) being most common. The majority of respondents indicated that they were unemployed/housewives (87.33%), with small percentages reporting farming (2.8%) and light trade/shop ownership (4.7%) as forms of employment. Nearly all (94.58%) respondents noted Islam as their primary religion.

Table 2 includes the number of participants who were exposed to NNCC interventions, including media, IPC, and a combination of the two. Nearly half (48.15%) of the respondents reported exposure to only NNCC media interventions, while only 3.69% of the respondents were exposed to only the IPC intervention. Approximately 18% of respondents reported exposure to both media and IPC.

Detailed knowledge and behavior frequencies are included in Table 3. Nearly half (47.75%) of the respondents reported high knowledge, and 33.62% reported medium knowledge of the risks of open defecation. Respondents’ knowledge of the benefits of handwashing was lower, with 14.94% and 37.72% reporting high knowledge and medium knowledge, respectively. Almost 70% of the respondents reported engaging in appropriate defecation behaviors. Only 4.79% of the sample reported washing hands at all critical times, and a larger proportion, 7.26%, reported never washing hands.

Table 4 includes results of the linear regression models testing for associations between exposure to the NNCC interventions and knowledge of defecation and handwashing. Exposure to a combination of both media and IPC (*p* < 0.001) and exposure to only media (*p* < 0.001) were associated with increased knowledge of defecation. Exposure to a combination of both media and IPC was associated with an increased knowledge of handwashing (*p* < 0.001), while there was no statistically significant association with exposure to only media or to only IPC.

Table 5 includes results of the logistic and linear regression models testing for associations between exposure to the NNCC interventions and appropriate defecation and handwashing behaviors. Exposure to only media was associated with appropriate defecation behavior (OR 1.539, CI 1.173–2.019), while there was no statistically significant association with exposure to only IPC or to both media and IPC. Exposure to both media and IPC (*p* < 0.001) and exposure to only IPC (*p* < 0.05) were associated with better handwashing behaviors while there was no statistically significant association with exposure to only media.

## 4. Discussion

The purpose of this study was to determine if exposure to either media, interpersonal communication (IPC), and a combination of both in a large social and behavior change campaign was associated with increases in WASH knowledge and behaviors among mothers with children under the age of two. The results show that a combination of both media and IPC interventions was closely associated with improved knowledge about both defecation and handwashing, while exposure to media alone was associated with knowledge about defecation but not about handwashing.

Findings from this study are consistent with other studies that have shown a positive impact of media on child health practices. For example, a systematic review of mass media’s impact on child health found that among 32 studies deemed methodologically rigorous, 26 showed a positive impact of mass media on self-reported health behaviors [21]. Similarly, a review of a national handwashing campaign in Ghana found the intervention reached over 80% of the population and noted that a combination of radio and television messaging had greater impacts on knowledge and behavior than community events [22].

Findings indicate that media messages like television, radio, and social media platforms implemented in the NNCC may best be suited for addressing or promoting knowledge and behavior related to defecation. One explanation for this finding may be the intimate nature of defecation and relative social discomfort with discussing defecation in a face-to-face setting. It may also be that open defecation, although relatively common in rural Indonesia, is nonetheless stigmatized and is thus less comfortable for both community health workers and mothers to discuss openly. In this case, media interventions offer a potential level of privacy at point of exposure that would be difficult, if not impossible, to achieve with IPC.

Mass media has the potential to extend the reach of health messaging across diffused populations like Indonesia, where 261 million residents are dispersed across 17,000 islands and 75,000 villages [23]. The media approach used in the NNCC was inclusive of social media, which may additionally help to reinforce the status of frontline health workers as educated, trusted, and accessible to community members [23]. Given the widespread lack of knowledge about stunting in rural Indonesia, the use of multiple media channels in the NNCC might have extended campaign coverage and may have encouraged behavior change. Wakefield, Loken, and Hornik [24] credit the effectiveness of mass media for promoting health behaviors to several key factors, including (1) removing emotional or cognitive obstacles to change, (2) helping people to either adopt healthy social norms or recognize unhealthy social norms, (3) associating valued emotions with achieving change, and (4) strengthening intentions for achieving behavior change. While the current study explored associations between media exposure and WASH-related knowledge and behaviors, future studies may continue to explore and verify the why and how of media impacts.

Exposure to only NNCC IPC interventions in this study was associated with one behavioral indicator for handwashing, while media-only exposure was not associated with any handwashing measure. It may be that IPC interventions, including the women’s groups, maternal health classes, and Posyandu services for women utilized in the NNCC, are best suited for addressing or promoting knowledge and behavior related to handwashing. Perhaps handwashing instruction, including modeling and practicing proper technique, is best accomplished through interpersonal communication and in settings where both knowledge of, and practice with, this important behavior can be accomplished. It is noted that standard approaches to WASH promotion, including face-to-face instructions, focused on educating individuals on the value of handwashing with soap following critical incidents, have been shown to have limited long-term impact on behavior change [25,26,27,28,29]. An exception to this finding is a small IPC intervention where Indonesian mothers were taught hand hygiene by community health workers and were provided free soap. A two-year follow up found that nearly 80% of participants were still using hand soap, despite the fact that it was no longer provided to them free of charge [30]. In this case, the face-to-face education provided by trusted community health workers may have made the difference. Greenland et al. note the effectiveness of IPC interventions increases when new mothers are targeted, and the information comes from trusted others, primarily grandmothers and midwives [31]. The current study is unable to measure the long-term impact of NNCC IPC strategies on WASH behaviors, yet the inclusion of women’s group instruction, maternal health classes, and Posyandu services provided by midwives provides hope that the significant impact of IPC identified in this study may be long-lasting.

The results from this study should be considered with several key limitations. This study was cross-sectional, and, therefore, unable to demonstrate a causal relationship between exposure to NNCC interventions and knowledge or behavior of study variables. Second, the study lacked sufficient power to be able to detect small effects, particularly for those variables with insufficient numbers of responses. In particular, limited numbers of participants exposed to IPC interventions may have prevented additional significant findings. Third, much of the data analyzed resulted from participant’s self-report and recall of exposure to NNCC programming and are therefore susceptible to error. Fourth, participants came from three rural regions of Indonesia, which are not representative of the country as a whole, greatly limiting the ability to generalize results. Finally, the temporal variance of exposure to NNCC programming and timing of the subsequent data collection may have impacted participant recall and impacted the study results. However, the research is still highly relevant as reaching rural communities is integral to the Sustainable Development Goal to achieve access to adequate and equitable sanitation and hygiene for all and end open defecation”. Finally, this study did not use an asset index to measure poverty, which is traditionally used as an indicator of wealth in developing settings. Since the indicators necessary to construct such an index were not included in the study survey, a measure of total household income was used instead.

## 5. Conclusions

In conclusion, analysis of exposure to media and IPC interventions included in the NNCC show that exposure to both IPC and media interventions were associated with improved knowledge and behaviors related to WASH. These findings demonstrate the value of communications campaigns that use mass media coupled with IPC to improve WASH knowledge and behavior. Media campaigns may be helpful when addressing knowledge and behaviors for appropriated defecation, while IPC interventions appear well-suited for promoting handwashing behavior.

## Figures and Tables

**Table 1 ijerph-17-03727-t001:** Participant demographics, *N* = 1734.

Demographics	*N* (%)/Mean (SD)
	Mother
Mean Age	27.72 (5.96)
Education	
None	97 (5.59%)
Primary School	670 (38.64%)
Junior High School	423 (24.39%)
Senior High School	434 (25.03%)
Tertiary Education	110 (6.34%)
Occupation	
Unemployed/Housewife	1468 (87.33%)
Farmer	47 (2.80%)
Light traders/Shop Owner	79 (4.70%)
Other	4 (0.24%)
Daily/Labor Worker	11 (87.33%)
Fisherman	6 (0.36%)
Private Employee	25 (1.49%)
Religion	
Islam	1640 (94.58%)
Other	94 (5.42%)
Mean Total Household Income (IDR)	1580, 2,193,594.18 (1,944,608.71)

Note: *N* is the number of respondents in each category.

**Table 2 ijerph-17-03727-t002:** Exposure to intervention: Media and IPC.

Exposure	*N* (%)
No exposure	522 (30.10%)
Exposure to both media and IPC	313 (18.05%)
Exposure to only media	835 (48.15%)
Exposure to only IPC	64 (3.69%)

Note: *N* is the number of respondents in each category.

**Table 3 ijerph-17-03727-t003:** Knowledge and behavior frequencies.

Variable	*N* (%)
**Knowledge**	
Knowledge of risks of open defecation	
No knowledge	35 (2.02%)
Low	288 (16.61%)
Medium	583 (33.62%)
High	828 (47.75%)
Knowledge of benefits of proper handwashing	
No knowledge	204 (11.76%)
Low	617 (35.58%)
Medium	654 (37.72%)
High	259 (14.94%)
**Behavior**	
Appropriate defecation behavior ^a^	1212 (69.9%)
No handwashing	126 (7.27%)
Washed hands at one critical time	286 (16.49%)
Washed hands at two critical times	443 (25.55%)
Washed hands at three critical times	490 (28.26%)
Washed hands at four critical times	306 (17.65%)
Washed hands at five critical times	83 (4.79%)

Note: *N* is the number of respondents in each category; ^a^ Dichotomous variable (appropriate vs. inappropriate).

**Table 4 ijerph-17-03727-t004:** Regression analysis with knowledge for defecation and handwashing.

Exposure	Adjusted Beta Estimate (*p*-Value) ^a^
**Knowledge of Defecation Model**	
Exposure to both media and IPC	0.402 (<0.001) ***
Exposure to only media	0.198 (<0.001) ***
Exposure to only IPC	0.133 (0.193)
**Knowledge of handwashing model**	
Exposure to both media and IPC	0.340 (<0.001) ***
Exposure to only media	0.057 (0.253)
Exposure to only IPC	0.131 (0.258)

^a^ Linear regression model; Reference group = no exposure; Each model adjusted for mother’s education, mother’s age, and total household income. *** *p* < 0.001.

**Table 5 ijerph-17-03727-t005:** Regression analysis with behavior for defecation and handwashing.

Exposure	Adjusted Beta Estimate (*p*-Value) ^a^	Adjusted OR (95% CI) ^b^
**Appropriate defecation model**		
Exposure to both media and IPC	-	0.780 (0.566–1.101)
Exposure to only media	-	1.539 (1.173–2.019) *
Exposure to only IPC	-	0.573 (0.322–1.017)
**Handwashing at critical times model**		
Exposure to both media and IPC	0.349 (<0.001) ***	-
Exposure to only media	0.0276 (0.543)	-
Exposure to only IPC	0.2693 (0.009) **	-

^a^ Linear regression model; ^b^ Logistic regression model; Reference group = no exposure; OR = odds ratio; 95% CI = confidence interval; Each model adjusted for mother’s education, mother’s age, and total household income. * *p* < 0.05, ** *p* < 0.01, *** *p* < 0.001.

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
