# Peer review of "A National Communication Campaign in Indonesia Is Associated with Improved WASH-Related Knowledge and Behaviors in Indonesian Mothers"

_ijerph, 2020, doi:10.3390/ijerph17103727_

Round 1

Reviewer 1 Report

Paper overall is good with very few grammatical issues. However, there are some concerns regarding same text Lines 206-208, 208-211 and 222-227. The discussion areas in this paper are the same thoughts with exception of a comma or transition word. Please the the following sources for examination. I suggest making corrections to these areas and citing these sources. 

https://www.mdpi.com/1660-4601/16/11/1963/htm?

https://www.scirp.org/journal/paperinformation.aspx?paperid=89488

For limitations, I would also include something about the time of day that the surveys were provided relative to the campaign itself. For example if the media campaign comes on at certain times of the day and the survey is being administered around that time such as right after this can influence results. Something else to consider in background is to share the time of day that WASH media, intervention or both was provided was provided and how that may have factored in and it's impact on dose effect.  

Author Response

Paper overall is good with very few grammatical issues. However, there are some concerns regarding same text Lines 206-208, 208-211  .

Thank you for this. There was indeed great redundancy in these sections. We have deleted an entire sentence and reworded another. We feel the manuscript flows much better now.

Change: For example, a systematic review of mass media’s impact on child health found that among 32 studies deemed methodologically rigorous, 26 showed a positive impact of mass media on self-reported health behaviors [21]. Similarly, a review of a national handwashing campaign in Ghana found the intervention reached over 80% of the population and noted that a combination of radio and television messaging had greater impacts on knowledge and behavior than community events [22].

Comment on lines 222-227,

Thank you for this. There was indeed far too much self-plagarism from our previous work in this paragraph. We have reworded each sentence and cited our previous paper on the NNCC in Indonesia.

Change: Mass media has the potential to extend the reach of health messaging across diffused populations like Indonesia where 261 million residents are dispersed across 17,000 islands and 75,000 villages [23]. The media approach used in the NNCC was inclusive of social media which may additionally help to reinforce the status of frontline health workers as educated, trusted, and accessible to community members [23]. Given the widespread lack of knowledge about stunting in rural Indonesia, the use of multiple media channels in the NNCC might have extended campaign coverage and may have encouraged behavior change.

The discussion areas in this paper are the same thoughts with exception of a comma or transition word. Please the the following sources for examination. I suggest making corrections to these areas and citing these sources.

https://www.mdpi.com/1660-4601/16/11/1963/htm?

https://www.scirp.org/journal/paperinformation.aspx?paperid=89488

For limitations, I would also include something about the time of day that the surveys were provided relative to the campaign itself. For example if the media campaign comes on at certain times of the day and the survey is being administered around that time such as right after this can influence results. Something else to consider in background is to share the time of day that WASH media, intervention or both was provided was provided and how that may have factored in and it's impact on dose effect.

Thank you! We have added the temporal variation to program exposure and data collection. We have not addressed the timing of programming in the Background section as suggested because we do not have that level of detail for media programming or participant exposure.

Change: Finally, temporal variance of exposure to NNCC programming and timing of the subsequent data collection may have impacted participant recall and impacted study results.  

Reviewer 2 Report

Thank you for doing this important work. WASH behaviors are an important topic. I hope that the following points will help you to strengthen your work before publication.

Material and Methods

The interventions carried out also include nutritional assessment? Why is there no nutrition related data in the results? It seems that the nutrition-related behaviors do not appear in this paper.

Discussion

Pag 6. Line 207. I don’t understand the sentence “including nutrition-related behaviors”.  This paper does not discuss nutritional habits, only WASH knowledge and behaviors

Author Response

Thank you for doing this important work. WASH behaviors are an important topic. I hope that the following points will help you to strengthen your work before publication

Comment #1: Thank you. The NNCC was a large effort consisting of many objectives. This manuscript only addressed WASH-related findings. The impact of NNCC nutrition-related objectives has been reported in other manuscripts.

Material and Methods

The interventions carried out also include nutritional assessment? Why is there no nutrition related data in the results? It seems that the nutrition-related behaviors do not appear in this paper.

Comment #2: Thank you for this comment. We can see how this sentence can be confusing and have removed “including nutrition-related behaviors” from the sentence.

Discussion

Pag 6. Line 207. I don’t understand the sentence “including nutrition-related behaviors”.  This paper does not discuss nutritional habits, only WASH knowledge and behaviors

Change made to Page 6, line 207: "For example, a systematic review of mass media’s impact on child health found that among 32 studies deemed methodologically rigorous, 26 showed a positive impact of mass media on self-reported health behaviors [21]."